# Effects of an isoenergetic low Glycaemic Index (GI) diet on liver fat accumulation and gut microbiota composition in patients with non-alcoholic fatty liver disease (NAFLD): a study protocol of an efficacy mechanism evaluation

Amina Al-Awadi [1,2,3] Jane Grove,[1,2] Moira Taylor,[4] Ana Valdes,[2,5] Amrita Vijay,[5] Stephen Bawden,[2,6] Penny Gowland,[2,6] Guruprasad Aithal[1,2]

For numbered affiliations see end of article.

**Correspondence to**
Professor Guruprasad Aithal; mszag3@exmail.nottingham.ac.uk

## ABSTRACT

**Introduction** A Low Glycaemic Index (LGI) diet is a proposed lifestyle intervention in non-alcoholic fatty liver diseases (NAFLD) which is designed to reduce circulating blood glucose levels, hepatic glucose influx, insulin resistance and de novo lipogenesis. A significant reduction in liver fat content through following a 1-week LGI diet has been reported in healthy volunteers. Changes in dietary fat and carbohydrates have also been shown to alter gut microbiota composition and lead to hepatic steatosis through the gut-liver axis. There are no available trials examining the effects of an LGI diet on liver fat accumulation in patients with NAFLD; nor has the impact of consuming an LGI diet on gut microbiota composition been studied in this population. The aim of this trial is to investigate the effects of LGI diet consumption on liver fat content and its effects on gut microbiota composition in participants with NAFLD compared with a High Glycaemic Index (HGI) control diet.

**Methods and analysis** A 2×2 cross-over randomised mechanistic dietary trial will allocate 16 participants with NAFLD to a 2-week either HGI or LGI diet followed by a 4-week wash-out period and then the LGI or HGI diet, alternative to that followed in the first 2 weeks. Baseline and postintervention (four visits) outcome measures will be collected to assess liver fat content (using MRI/S and controlled attenuation parameter-FibroScan), gut microbiota composition (using 16S RNA analysis) and blood biomarkers including glycaemic, insulinaemic, liver, lipid and haematological profiles, gut hormones levels and short-chain fatty acids.

**Ethics and dissemination** Study protocol has been approved by the ethics committees of The University of Nottingham and East Midlands Nottingham-2 Research Ethics Committee (REC reference 19/EM/0291). Data from this trial will be used as part of a Philosophy Doctorate thesis. Publications will be in peer-reviewed journals.

**Trial registration number** NCT04415632.

## Strengths and limitations of this study

► A carefully controlled human Low Glycaemic Index diet intervention to reduce liver fat in non-alcoholic fatty liver diseases (NAFLD) using cross-over study deign.

► The first NAFLD patient study that explores the mechanistic effects of dietary glycaemic index on liver fat accumulation using advanced MR protocols.

► Uses the controlled attenuation parameter as a biomarker to evaluate efficacy of diet intervention.

► Findings from small sample size may not be generalisable to all patients with NAFLD so will require replication in larger long-term randomised controlled trial.

► The 2-week time frame of the intervention, especially in light of isoenergetic diet, may not be sufficient to observe significant changes in liver fat.

## INTRODUCTION

Non-alcoholic fatty liver disease (NAFLD) is a growing global health concern increasing in parallel with the worsening epidemics of obesity and diabetes.[1] [2] Recent reviews indicated that one billion people worldwide have NAFLD (estimated prevalence 25.4%).[3] NAFLD is characterised by the fat accumulation in hepatocytes exceeding 5% of liver weight and encompasses a spectrum of disease severities. Common clinical risk factors that are highly associated with NAFLD include obesity, insulin resistance, type 2 diabetes and metabolic syndrome.[3] [4] NAFLD is recognised as the hepatic manifestation of metabolic syndrome.[5–7]

Western dietary habits occurring concurrently with a sedentary lifestyle and genetic factors are associated with insulin resistance, adipocyte

BMJ

proliferation and gut microbiota alteration which may be implicated in NAFLD.[8] Dietary habits, hormones secreted from the adipose tissue, gut microbiota composition and genetic factors are all implicated in the pathogenesis.

To date, there are no drugs with regulatory approval for treating NAFLD, and lifestyle changes including modification of dietary patterns together with increased physical activity is the initial step to treat this disease.[9] Dietary approaches beneficial in NAFLD management include reductions in saturated and trans fatty acids, total carbohydrates and animal-based protein intake and increases in the intake of polyunsaturated fatty acids, monounsaturated fatty acids, plant-based proteins and antioxidants.[4] Furthermore, evidence from a systematic review and meta-analysis of randomised controlled trials (RCTs) on patients with NAFLD has shown that lifestyle-induced weight loss of ≥7% improved histological disease activity, as well as cardiometabolic risk profile, in less than 50% of the study population (n=373 patients).[10] A recent systematic review has concluded that despite the abundance of interventional studies on dietary management in NAFLD, there is still a lack of evidence-based specific recommendations, other than that weight loss may benefit some; and few studies have measured liver-related outcomes.[11] Moreover, the dietary effects on liver-related outcomes that are mediated by gut microbiome have not been studied in these patients.[12]

More recently, the Glycaemic Index (GI) and glycaemic load (GL), of diet; has been proposed as an important predictor of NAFLD progression.[4 13] The GI of a food ranks the glycaemic response under conditions in which the quantity of carbohydrate is controlled,[14 15] whereas the GL additionally considers portion size.[16]

NAFLD patients have been found to consume a diet high in High GI (HGI) foods[4] which are correlated with insulin resistance, metabolic dysfunction,[4] and liver dysfunction.[17] In addition, HGI dietary habits have been significantly associated with high-grade liver steatosis in patients with insulin resistance[18] while limiting HGI foods is beneficial in NAFLD.[19 20] Further, previous pilot study showed that only 1 week of consuming HGI diet caused significant increase in hepatic stores of fat when compared with isocaloric Low GI (LGI) diet consumption.[21] A recent review concluded that both low GI and GL diets resulted in significant reductions in hepatic fat mass in a cohort of 269 participants.[16] However, these effects were not attributed to the LGI diet alone and were confounded with other lifestyle interventions.[16]

Evidence from animal and human studies also showed that diet is the most important modifying factor in the development of human gut microbiome.[22–24] Dietary alterations can rapidly change the human gut microbiota composition in relatively short time periods[25–27] and in as little as 24 hours.[28–30] Certain dietary factors which modify gut microbiota have been associated with a number of diseases including obesity[31] metabolic syndrome,[22 32 33] type 2 diabetes,[34] immunological dysfunctions.[35] Such dysbiosis in diet-induced NAFLD has been shown to stimulate hepatic fat disposition and promote NAFLD progression to non-alcoholic steatohepatitis (NASH).[36 37] This is suggested to be mediated by modulation of gut permeability, dietary choline metabolism, bile acid metabolism, endogenous ethanol production and immune balance.[36 37] Gut microbiota modulated by diet can increase the energy extraction from the breakdown of non-absorbable polysaccharides with the production of monosaccharides and short chain fatty acids (SCFAs) which are substrates for hepatic de novo lipogenesis and carbohydrate biosynthesis.[38] Certain SCFAs have also been implicated in improving glucose and lipid metabolism beneficial to the host.[39] Therefore, there is a potential to modulate gut microbiota through dietary changes as a means of NAFLD prevention or alleviation.[11 40]

We describe here our study protocol designed to assess the effects of a specific defined GI dietary manipulation[21] on liver fat content and its effects on gut microbiota composition in participants with NAFLD compared with a higher GI diet as a control diet. The diets will be adjusted to each participant to meet their energy requirements and the macronutrient composition of the diets will be carefully controlled ensuring that differences in GL are due to differences in GI.

## OBJECTIVES
### Primary objective
1. To investigate the effects of defined LGI versus HGI diet on liver fat accumulation in participants with NAFLD following a 2-week intervention period while controlling for the energy consumption and amount of carbohydrate intake.

### Secondary objectives
1. To investigate the effects of LGI versus HGI diet on gut microbiota composition and its correlations with liver outcomes in participants with NAFLD following a 2-week intervention period.
2. To investigate the changes in blood metabolic biomarkers (glycaemic, insulinaemic, liver, lipid, haematological profiles, gut hormones and plasma SCFAs) between LGI and HGI diet groups following a 2-week intervention in participants with NAFLD.
3. To evaluate controlled attenuation parameter (CAP) as a tool to test the efficacy of a dietary intervention by comparing its performance characteristics with that of MR Spectroscopy (MRS) as a gold standard.
4. To validate the European Prospective Investigation of Cancer Food Frequency Questionnaire (EPIC-FFQ) in Nottingham NAFLD patients against food diary.

## METHODS AND ANALYSIS
### Trial design
This is a randomised 2×2 cross-over mechanistic trial including two arms, each of 2 weeks duration comparing an LGI diet and a control HGI diet based on that defined by Bawden et al[21] and Morgan et al.[41] The study design is illustrated in figure 1.

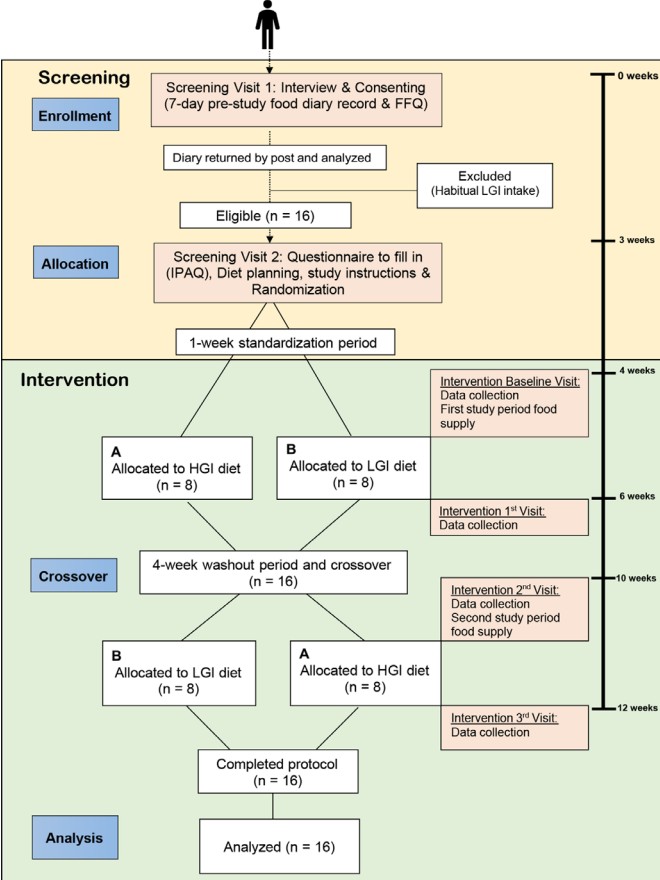

**Figure 1** Schematic diagram for cross-over trial of LGI diet versus HGI diet effects on liver fat content in participants with NAFLD. FFQ, Food Frequency Questionnaire; HGI, High Glycaemic Index; IPAQ, International Physical Activity Questionnaire; LGI, Low Glycaemic Index; NAFLD, non-alcoholic fatty liver diseases.

**Box 1   Eligibility criteria**

### Inclusion criteria

► Adult males and females aged from 18 to 65 years (balanced number).
► Detected non-alcoholic fatty liver diseases (NAFLD) by controlled attenuation parameter (CAP)-FibroScan >288 dB/m or by MRI-proton density fat fraction >5% fat of liver weight.
► Body mass index ≥25 kg/m$^2$.
► Abdominal obesity (waist circumference >102 cm for males and >88 cm for females).
► Have current moderate to high Glycaemic Index diet intake of ≥60 (Assessed from a completed 7-day day food diary).
► Available to give informed consent.
► Available to undergo MRI/spectroscopy and CAP-FibroScan.

### Exclusion criteria

► Current smokers and excessive alcohol drinkers (>14 units/week).
► Perimenopausal (irregular periods) women.
► Participants with other liver abnormalities.
► Participants with history of gastrointestinal surgeries, depression, eating disorders or difficulties.
► Participants using pharmacological agents for obesity or NAFLD.
► Participants with type 1 diabetes.
► Participants with type 2 diabetes on second line medications (eg, glucagon-like peptide-1 analogues).
► Participation in any other trial in the last 3 months.
► Participants on any special diets (eg, vegetarians).
► Intolerance to foods included in the diet plan.

## Public and patient involvement

We collaborated with the National Institute of Health Research (NIHR) Nottingham Biomedical Research Centre (BRC) patient advisory group, and we met seven patients who were interested in study participation. They expressed strong support for the research objectives and the recent meeting the group considered the 2-week cross-over intervention feasible and acceptable. Patient feedback informed the participant recruitment through information content and delivery methods. They also assisted in the study design to enhance participant compliance and commitment to the diet.

## Participants and recruitment

We aim to recruit 16 participants with NAFLD to complete both phases of 2×2 cross-over trial. The initial screening criteria for inclusion is a CAP score of >288 dB/m; this value correlates with MRI- Proton Density Fat Fraction (PDFF) estimation of >5% fat of liver weight which indicates that they have a fatty liver.[42]

NAFLD patients will be recruited either from the existing database of 'The Scarred Liver Project' in the UK who have given consent to be contacted to participate

in future research (approved by East Midlands-Leicester Research ethics committee ref: B/EM/0123) or from liver clinics at the Nottingham University Hospitals NHS Trust who had a FibroScan with CAP or participants from public (advert in local and social media). The eligibility criteria are summarised in box 1. All participants will provide a written informed consent by trained research team before they enter the trial. Potential participants will be free to withdraw at any time. In the event of their withdrawal, it will be explained that their data collected so far cannot be erased and we will seek consent to use the data in the final analyses where appropriate. In the event of a participant withdrawing or being unable to continue the intervention periods after randomisation, a replacement will be recruited who will be assigned to the same diet group to ensure a 1:1 ratio for the two diet groups. We will continue recruiting until 16 participants have fully completed the study.

## Screening process

Participants will be invited to attend an initial screening visit (figure 1). At this visit, participants will be consented and asked to complete a prescreening questionnaire (including taking height, weight and blood pressure measurements) to establish whether they meet the study criteria. Participants will be asked to fill out a 7-day food record and post it after 1 week of their initial screening visit. This is to avoid giving patients with NAFLD, who might be at risk of developing diabetes, a much higher

HGI diet than they habitually consume. Using 7-day food records, we will characterise the baseline habitual GI diet intake of all potential volunteers at screening period. Participants also will be required to complete a FFQ for validation against the 7-day food records. We will use EPIC-FFQ that was validated for use in Norfolk cohort.[43] After completion, participant's eligibility will be reassessed based on analysis of diet information using Nutritics software (Nutritics, Dublin, Ireland, 2018). Those with a habitual LGI diet intake (GI <60) will be excluded from the study. Only patients who have a moderate to HGI diet intake (GI ≥60) will be invited to attend the next screening preparation visit.

Then at next screening visit (visit 2), the participants will be asked to fill out International Physical Activity Questionnaire (IPAQ).[44] A trained dietitian will instruct participants on how to measure study food provided using measuring jug and spoons and how to fill in study diet compliance sheets. They will be instructed on stool collection and preintervention visits requirements. Participants will be asked to avoid eating probiotics and/or prebiotics supplements or drinks for 1 week before the intervention period (standardisation week). All instructions will be provided to each participant in a form of Home Guide Booklet.

Each eligible participant will be assigned randomly to undergo either HGI diet (HGI group) or LGI group first and then crossover to either HGI or LGI diets.

### Randomisation and blinding

Enrolled participants who meet the eligibility criteria will be randomised to one of two groups which will dictate the order in which they receive the LGI or HGI diet. They will then receive the second diet depending on which diet they have first received. The random allocation sequence (AB or BA) will be generated by one of the trial co-ordinators; who will have no contact with the participants, using block randomisation by impartial biostatistics. The online randomisation package (http://randomization.com) will be used to allocate the sequences of the diet groups to the participants in equal numbers. Once the random number is generated, both trial investigators and participants are aware of the diet sequence (ie, whether the participant is going to start with LGI diet; A or HGI diet; B). The diets then will be delivered to the participant according to the sequence. The trial outcome measures will be determined by blinded researcher using codes on samples that will be sent for 16S ribosomal RNA (rRNA), blood marker and MRI/CAP-FibroScan analyses. The blinded allocation will be provided to the study statistician at the end of the study and once the analysis has been completed, the study statistician will be provided with the unblinded allocation.

### Study protocol

Following the screening phase and standardisation week, each participant will experience two, 2-week intervention phases, with a 4-week washout period between interventions. Before and after each intervention period they will attend the Imaging Centre (Sir Peter Mansfield Imaging Centre) for MRI scan measurements and then will attend the Biomedical Research Centre for FibroScan, and blood and stool samples collection. During each intervention phase each participant will be required to consume all the food that has been provided, fill in dietary compliance sheets and complete Visual Analogue Scales (VAS) relating to subjective appetite ratings.

### Design of the dietary interventions

Energy expenditure will be estimated for each participant using the individual's basal metabolic rate calculated from Henry-modified Schofield formula[45] and multiplied by an activity factor based on the IPAQ score physical activity levels. A menu, consisting of meals for 3 days personalised for energy requirement, will be devised (these meals are repeated over 2 weeks) by adjusting a template according to their estimated energy requirement in order to minimise weight change during the intervention phase. The diet consists of normal food closely resembling that described by Bawden et al study.[21] A diet template of 2000 kcal/day with energy proportions distributed as 55% from carbohydrates, 15% from protein and 30% from fat[46] are designed and food quantities will be modified to supply the energy requirements for each participant. Three-day cycle menu plans are designed for each diet (LGI/ HGI). Table 1 shows example template menus for both HGI (a) and LGI (b) diets.

The GI values of commercial food items are generated from that reported by Henry et al,[47] as well as Nutritics software databases (Nutritics). The dietary GI of the menu is calculated from GI proportional contribution to total carbohydrate using the equation: $GI = \sum (C_{food}/C_{total}) \times GI_{food}$; where $C_{food}$ is the amount of carbohydrate contained in each digested food (g), $C_{total}$ is the total carbohydrate in the meal and is the reported GI of the food.[21] The GL of the diet is calculated based on the dietary GI and the amount of carbohydrate provided (the same in both diets) using the following equation: $GL = (GI_{food} \times C_{food})/100$; where GI is the GI of each digested food and food is the amount of carbohydrate in the food.[21] Food composition analysis and evaluation were performed using Nutritics software (Nutritics, Dublin, Ireland, 2018).

Participants will be provided with the food at the start of each 2-week diet period and will be clearly instructed on safe food storage and reheating. They will be required to complete dietary compliance sheet on a daily basis. Instructions on dietary compliance will be given at each study period and the dietary compliance records will be reviewed regularly during the intervention to assess their compliance. For the first 1 week prior to the intervention (1-week standardisation), and during the 4-week wash-out period, participants will be asked to avoid eating probiotics, yoghurt and cultured milk drink; if any, to standardise their gut microbiota and avoid any carry-over effects as shown in previous studies.[48 49] Meal-timing will be standardised among all participants as much as

**Table 1** Illustration of HGI (A) and LGI (B) diet templates for use in the study

| | Weight (g) | Energy (kcal) | Carbohydrates (g) | Protein (g) | Fat (g) | Fibre (g) | GI | GL |
|---|---|---|---|---|---|---|---|---|
| **HGI diet sample (A)** | | | | | | | | |
| **Breakfast** | | | | | | | | |
| Sultana Bran Cereal | 90 | 268 | 58 | 6.6 | 1.3 | 10.1 | 90 | 52.2 |
| Skimmed milk | 180 | 63 | 8.3 | 6.3 | 0.54 | 0 | 48 | 4 |
| Roasted peanuts | 15 | 90 | 2.3 | 3.7 | 7.4 | 1.3 | 14 | 0.32 |
| Glucose | 10 | 40 | 10 | 0 | 0 | 0 | 100 | 10 |
| Water | 170 | 0 | 0 | 0 | 0 | 0 | 0 | 0 |
| Meal total | | 461 | 78 | 16.6 | 9.2 | 11.4 | 85.3 | 66.5 |
| **Lunch** | | | | | | | | |
| Margarine | 25 | 102 | 0.13 | 0.05 | 11.3 | 0 | 0 | 0 |
| Fruit loaf | 144 | 395 | 76 | 11.4 | 5.2 | 4.6 | 90 | 68.4 |
| Roasted turkey slice | 42 | 47 | 0.67 | 10 | 0.5 | 0.12 | 0 | 0 |
| Cheddar cheese slice | 25 | 104 | 0.03 | 6.4 | 8.7 | 0 | 0 | 0 |
| Strawberry yoghurt | 125 | 88 | 16.3 | 5.6 | 0 | 0.38 | 85 | 13.855 |
| Meal total | | 735 | 93 | 33.4 | 25.7 | 5.1 | 88.44 | 82.255 |
| **Dinner** | | | | | | | | |
| Cottage pie (ready meal) | 400 | 441 | 44 | 18 | 21.6 | 4.8 | 66 | 29.04 |
| Low fat strawberry yoghurt | 125 | 88 | 16.3 | 5.6 | 0 | 0.38 | 85 | 13.855 |
| Water | 170 | 0 | 0 | 0 | 0 | 0 | 0 | 0 |
| Glucose | 10 | 40 | 10 | 0 | 0 | 0 | 100 | 10 |
| Mars bar | 58 | 237 | 37 | 2.4 | 8.9 | 0.87 | 65 | 24.05 |
| Meal total | | 806 | 107 | 26 | 30.5 | 6 | 72.69 | 76.954 |
| Menu total | | 2002 Kcal | 278 g (55.5%) | 76 g (15.2%) | 65 g (29.3%) | 22 g | 82.1 | 225.706 |
| **LGI diet sample (B)** | | | | | | | | |
| **Breakfast** | | | | | | | | |
| All Bran cereal | 90 | 239 | 42 | 11.2 | 3.1 | 22.1 | 43 | 18.06 |
| Semi-skimmed milk | 160 | 76 | 7.2 | 5.6 | 2.7 | 0 | 25 | 1.8 |
| Orange juice | 250 | 96 | 22 | 1.5 | 0.25 | 0 | 53 | 11.66 |
| Meal total | | 410 | 71 | 18.2 | 6 | 22.1 | 44.39 | 31.52 |
| **Lunch** | | | | | | | | |
| Pumpernickel bread | 180 | 408 | 74 | 15.7 | 5.6 | 11.7 | 41 | 30.34 |
| Black cherry yoghurt | 150 | 209 | 24.2 | 4.8 | 10.4 | 0.75 | 17 | 4.114 |
| Soft cheese | 51 | 129 | 1.5 | 2.7 | 12.4 | 0 | 34 | 0.51 |
| Meal total | | 745 | 99 | 23.2 | 28.4 | 12.5 | 35.32 | 34.96 |
| **Dinner** | | | | | | | | |
| Lasagne (ready meal) | 290 | 413 | 42 | 21.5 | 17.7 | 4.9 | 36 | 15.12 |
| Apple | 174 | 92 | 20 | 1 | 0.87 | 2.1 | 38 | 7.6 |
| Semi-skimmed milk | 150 | 71 | 6.8 | 5.3 | 2.6 | 0 | 25 | 1.7 |
| Fruit and nuts mix | 60 | 267 | 24.6 | 8.4 | 15 | 3.9 | 15 | 3.69 |
| Meal total | | 843 | 93 | 36.2 | 36.1 | 10.9 | 30.22 | 28.11 |
| Menu total | | 1999 kcal | 264 g (52.7%) | 78 g (15.5%) | 71 g (31.8%) | 46 g | 36.64 | 94.594 |

GI, Glycaemic Index; GL, glycaemic load; HGI, High Glycaemic Index; LGI, Low Glycaemic Index.

possible during the intervention as well as during the standardisation period.

## Data collection and analyses of outcome measures

The timing and frequency of study measurements are depicted in figure 1. Participants will be asked to come fasted (a minimum of 8 hours) for each laboratory visit (pre and post each intervention period) and the following measurements will be made:

1. Anthropometric measurements: Participant height (cm), weight (kg), waist circumference (cm), body mass index (BMI) $(kg/m^2)$ and body composition with bladder voided using bioelectrical impedance analysis Analisi Composizione Corporea (BIA-ACC)

appliance model BIA00A; (BioTekna, Marcon VE, Italy).

2. Liver fat assessment: Hepatic fat fraction and comparison lipid composition will be measured using a Philips 3T Acheiva MRI scanner with 32 channel receive XL torso coil and advanced lipid composition analysis on a Philips 7T Acheiva scanner with eight channel multitransmit body coil. MRS will be acquired from a 20×20×20 mm voxel within the lower right lobe of the liver using Stimulated Echo Acquisition Mode localisation to assess hepatic fat fraction and composition. Spectra will be acquired over four echo times (20, 30, 40, 80) to correct for $T_2$ relaxation over five breath-holds. Water will be used as a reference peak, and water suppression used to analyse signal from lipids. Spectra will be post processed (phase corrected, line-broadened, frequency aligned and averaged) as described previously,[50] and then fitted according to known lipid signal positions.[51] $T_2$ corrected PDFF will be calculated as PDFF=FAT/(FAT +WATER) and converted to true mass fat fraction using known tissue values.[52] Lipid composition will be estimated using individual peak amplitudes.[53] Subcutaneous and visceral fat volumes will be determined using gradient echo MRI.[54]

   Liver fat levels will also be assessed with the CAP scored using FibroScan, which has previously been evaluated against MRS[55] as well as liver biopsy[56] in cross-sectional cohorts. Although MRI-PDFF more accurately classifies steatosis,[57] portability, accessibility and costs favour CAP associated with transient elastography. However, CAP has not been evaluated as a marker of efficacy of intervention. For the purpose of validation, participants will undergo a CAP-FibroScan at each intervention visit to quantify their liver fat levels.

3. Hepatic ATP flux: Rate of mitochondrial ATP turnover in the liver will be assessed using a Philips 3T MRI scanner with a Philips Phosphorous (31P) single loop surface coil. 31P MRS will be acquired from the liver using a 60 mm thick slice to avoid contamination from abdominal muscle. A progressive saturation scan protocol (five spectra acquired with varying length presaturation pulses) will be used to determine apparent T1.[58] Two 31P MR spectra will then be acquired with full saturation of gamma ATP (−2.5 ppm) and mirrored downfield of the inorganic phosphate peak (~14 ppm) respectively. An in-house fitting programme will be used for postprocessing spectra and measuring the saturation transfer to the inorganic phosphate peak and used to determine hepatic ATP flux rate constant. ATP concentration will also be determined from spectra and used to calculate final ATP flux rate values.[59]

4. Metabolic Biomarkers: Blood samples (not more than 80 mL at each visit) will be taken from fasted participants (12-hours fast) for analyses of metabolic analytes such as: glycaemic, insulinaemic, liver, lipid, haematological profiles, gut hormones (ghrelin, leptin, glucagon-like peptide-1 (GLP-1), glucagon) and plasma SCFAs. Although GLP-1 levels increase in response to meal ingestion, levels can be directly affected by presence of SCFAs, produced from fermentation of certain carbohydrates which may have been recently consumed, confounding the results. Instead, only fasting levels will be measured as short-term diets containing different carbohydrates types and different GI content have been found to affect fasting GLP-1 levels.[60] In addition, circulating plasma SCFAs levels will be measured as it has been shown to be strongly related to metabolic health.[61] Measurement of plasma SCFAs levels also overcomes the challenges associated with accurate quantification of volatile SCFAs molecules in faecal samples.[62]

   All biochemical parameters concentrations will be assessed by a central laboratory that uses standard methods. Standard operating procedures will be followed for all laboratory assays to ensure the high quality and reliability of our generated data. Routine blood tests will be analysed by the accredited laboratories. Samples will be stored in −80°C freezers until the study ends, under yearly service contract and monitoring systems. Analyses will be done according to assay manufacturer advice and completed by technicians and research staff who are formally trained or outsourced to collaborators or external suppliers where necessary. The insulin resistance will be estimated using homeostatic model assessment of insulin resistance (HOMA-IR). HOMA-IR is calculated according to the equation (HOMA-IR = (FPI×FPG)/22.5); where FPI is fasting plasma insulin, and FPG is fasting plasma glucose.[4]

5. Gut Microbiota Composition: Each participant will be asked to collect fresh stool sample in a standard laboratory container and bring it to the research centre with an ice pack for storage at −80°C. Stool samples will be processed using a standard commercial kit for (DNA) extraction.[63] Then, the gut microbiota composition will be assessed using 16S sequencing of rRNA. DNA will be diluted to 20 ng/μL for 16S rRNA amplification and sequencing. Water negative controls will be included from extraction, through PCR to sequencing and select samples were sequenced in duplicate for quality control. The V4 region of the 16S rRNA gene will be amplified using universal primers 355F (CCAGACTCCTACGGGAGGCAGC) and 806R ( GGACTACHVGGGTWTCTAAT). Amplified DNA will be sequenced on the MiSeq platform (Illumina). Read filtering and clustering will carried out using the MYcrobiota pipeline. Briefly, chimeric sequences will be filtered using the VSEARCH algorithm within Mothur, and reads will be clustered into operational taxonomic units (OTUs) using closed-reference clustering against the SILVA database v132 based on a 97% similarity. Diversity metrics (Shannon index, observed OTUs and Unweighted UniFrac) will be calculated by rarefying the OTU table down to 7000 sequences per sample 50 times and taking the average. These analyses will be carried out in QIIME V.2 (V.2018.11).[64 65]

6. Subjective Sensation Assessments: VAS scores will be used to assess the participant's subjective appetite (hunger and fullness) during the intervention. Participants will be provided with a booklet to record their subjective appetite before and after each meal during the intervention at selected days.

## Sample size calculation and justification

The sample size is calculated using nQuery based on previous completed study by Bawden *et al.*[21] Based on their pilot data, baseline liver fat fractions of both diet groups were 2.4±1%–1.2%, and after 1week were HGI (High GI)=4.7% ± 2% and LGI=1.6 ± 0.7%. Using a two-group t-test (cross-over analysis of variance (ANOVA)) between the post-diet values with a 0.05 two-sided significance level, this produces a sample size of n=8 in each diet sequence (a total sample size of 16). This sample size has an 80% power to detect a difference in means of 3.1 (the difference between a Treatment 1, $\mu_1$, of 4.7 and a Treatment 2, $\mu_2$, of 1.6) with the assumption that the crossover ANOVA sqrt (MSE) is 2.828 (the SD of differences, $\sigma$, is 4). Assuming that 40% of people would drop out from cross-over study design, it is expected that at least 24 participants would need to be recruited.

The effect from the pilot study by Bawden *et al* pilot study (n=7) was an average 1.6%±0.7% decrease in liver fat fraction after following a 1-week LGI diet (p<0.05). Given that baseline liver fat fraction was 2.4%, in order to see such change with 80% power (p<0.05) we have doubled the participant numbers to n=16 using nQuery software to detect statistically significant differences.[21] Our group has recently completed a study (ClinicalTrials.gov Identifier: NCT03844165) in which 84 participants with NAFLD followed either an LGI food enriched diet or their normal diet. Patients who changed their diet were found to have a significant decrease in liver fat after 16 weeks (from 20.4% to 18.8%) and had a statistically significant decrease in fasting glucose levels after the diet compared with the control group patients following their normal diet. This provides strong support to our hypothesis and demonstrates that a simple dietary change can have measurable impact on NAFLD symptoms in a short period. In another previous randomised crossover trial, metabolic changes were seen within 2weeks in a dietary intervention with 10 participants.[66] Based on these different approaches, we conclude that a sample size of 16 at the end of the study is sufficient. Assuming a 33% drop-out rate we estimate that it will be necessary to enrol n=24 NAFLD participants to achieve n=16 completing the two diet-interventions. This dropout rate has been reported in other dietary interventional studies.[67]

## Statistical analysis plan

Baseline characteristics of participants, outcome measures at baseline (liver fat content, blood biomarkers and gut microbial composition) at each period time point will be summarised by diet group using descriptive statistics. The differences between the two interventions (LGI and HGI

diet) will be tested by matched-pair Student's t-test. Two-way repeated measures ANOVA test will be used to evaluate the effects of diets and time sequences on liver fat content, as well as on gut microbial composition and blood biomarkers and/or interactions with liver fat content, gut microbial composition and blood biomarkers. Data will be presented as mean±SD and the significance of difference will be set at p<0.05. Any changes in the planned statistical methods will be documented in the trial report. Data input, cleaning and analysis will be conducted using Stata software package. All analyses will be conducted on University of Nottingham computers/laptops that are regularly backed up to University of Nottingham servers.

Microbiome analyses: OTUs with a relative abundance of <0.1% in every sample will be removed, and relative OTU abundances will be inverse normal transformed before further analysis. Associations between response to LGI diet and OTU abundance at genus level will be predicted using a general linear model adjusted for age, gender and BMI. OTUs will be assumed to be significantly associated with response to the dietary intervention with a value <0.05 after adjusting for false discovery rate.

## Data management

Trial data will be managed by study coordinator (AA-A) under the supervision of chief investigator (GA) and study statistician. The study coordinator shall carry out monitoring of trial data as an ongoing activity and perform annual site system audit. Trial data and evidence of monitoring and systems audits will be made available for inspection by REC as required. The sponsor has the right to take advice from the Trial Steering Committee and Data Monitoring Committee as appropriate for study discontinuation. All trial staff and investigators will endeavour to protect the rights of the trial's participants to privacy and informed consent, and will adhere to the Data Protection Act, 2018. Access to the data will be limited to the trial staff and investigators and relevant regulatory authorities and will be restricted by user identifiers and passwords (using a one-way encryption method). Computer held data including the trial database will be held securely and password protected. All data will be stored on a secure dedicated web server. Electronic data will be backed up every 24 hours to both local and remote media in encrypted format.

In compliance with the International Conference on Harmonisation Good Clinical Practice (Food and Drug Administration guideline) guidelines, regulations and in accordance with the University of Nottingham Research Code of Conduct and Research Ethics, the chief or local principal investigator will maintain all records and documents regarding the conduct of the study. These will be retained for at least 7 years or for longer if required. The Trial Master File and trial documents held by the Chief Investigator on behalf of the sponsor shall be finally archived at secure archive facilities at the University of Nottingham. This archive shall include all trial databases and associated meta-data encryption codes.

## Ethics and dissemination

Ethical approval was first obtained from the University of Nottingham, Medical School Research Ethics Committee. Then additional ethical approval was obtained from East Midlands Nottingham 2 Research Ethics Committee (The Old Chapel, Royal Standard Place, Nottingham). Within a 5-year time frame, the final anonymised trial data will be uploaded to the https://rdmc.nottingham.ac.uk public repository after consistency and quality have been verified by the project team and publication of the results. Data from this trial are intended to be presented at local and international conferences including those attended by clinicians and dietitians and will also be used as part of a Philosophy Doctorate thesis. Publications will be in peer-reviewed journals. No participant will be identified in any of these publications.

## DISCUSSION AND RESEARCH IMPLICATIONS

Building on previous findings from Morgan *et al*[41] and Bawden *et al*,[21] this study will investigate the effects of LGI versus HGI diets in participants diagnosed with NAFLD, over a 2-week period. While previous studies showed a significant impact of LGI on liver fat reduction and other metabolic biomarkers, questions remain on the effect in individuals with NAFLD and the longer-term impact. Due to the carefully controlled nature of the intervention, the sample size can be reduced, and the research used to probe more specifically into the direct effects of varying the type of carbohydrate intake. Many studies have shown that energy reduction has a profound effect on liver fat and metabolic health,[68] but these diets have been found to be unsustainable in a community setting.[10] Proof of the direct impact of varying a specific characteristic of one macronutrient without reducing the overall intake of the macronutrient consumed, or the overall energy consumed would provide a strong basis for further investigations on the sustainability of such diets.

This study takes advantage of advanced MR protocols which provide a powerful method for non-invasively assessing the effects of lifestyle interventions on metabolic health alongside other physiological measures. Biopsy measurements are uncomfortable for the patient carrying a number of significant risks and would be ethically inappropriate in otherwise healthy individuals. In addition, biopsy only provides a small sample from one region of the liver. MRI and MRS have been well validated[52 69] and allows for repeated tissue specific measurements of fat fraction with minimal risk to study volunteers, thus enabling robust longitudinal studies on the impact of nutrition on metabolic health and liver disease. Additionally, recent research has highlighted the potential significance of lipid composition in metabolic disorders and novel methodologies for measuring this using MRI and MRS have been developed.[53] These advanced techniques will be used to assess not only lipid accumulation but also the effect on saturated, unsaturated and polyunsaturated fat stores in the liver. Subcutaneous and visceral fat volumes will also be obtained alongside hepatic fat fraction during the scan session giving a more complete picture of abdominal lipid deposition.

31P MRS measurements of hepatic ATP flux provides a novel and exploratory assessment of the impact of lifestyle interventions on metabolism. Previous studies have shown the correlation of metabolic disorders such as NAFLD with impaired ATP turnover in the liver.[70] Using advanced in vivo saturation transfer MR protocols, this study will monitor these effects during an intervention and provide the first in-human study to explore mechanistic effects of diet on liver lipid accumulation. This will give a foundation for future research projects and help to advance the effort against the growing global pandemic of NAFLD and related disorders.

Outcomes from this study will provide evidence on the effects of an LGI diet consumption on measures of NAFLD and will serve as a complementary tool for reducing hepatic fat accumulation in NAFLD patients. It will also be the first to provide nutritional-related information about the effects of LGI diet on gut microbiota composition in this medical condition. Accordingly, trial findings will open the way for academic beneficiaries (including dietitians, hepatologists, diabetologists) to develop a novel strategy for NAFLD monitoring through understanding the mechanism behind the LGI dietary-related effects of specific gut microbiota and hepatic fat accumulation.

In addition, the validation findings for the use of CAP-FibroScan as a new tool for assessing hepatic fat levels in dietary interventions has the potential to enhance diagnostic and research capacity. Furthermore, the validation of Norfolk EPIC-FFQ in Nottingham population will permit its use in larger population-based studies.

If this study successfully reproduces the reduction in hepatic fat levels through LGI diet reported in healthy volunteers[21] in an NAFLD patient cohort, this will be a key step towards developing a new option for dietary intervention. The inclusion of LGI foods as a recommendation for NAFLD patients may have substantial beneficial impacts on society as well. It will first enhance the quality of life, health and well-being of NAFLD patients by reducing the burden of NAFLD progression to advanced liver diseases and more likely to be practical and thriving in the long term as it is the first non-calorie restriction approach. Second, it will contribute towards evidence-based policy making in GI food labelling and therapeutic dietary recommendations and will contribute to increase public awareness of dietary issues. The potential medical costs for a patient with NAFLD per year is between €354 and €1163 in Europe.[3] Thus, preventing NAFLD progression will reduce the burden on healthcare systems and clinical services as NAFLD patients have high-risk liver-related morbidities and metabolic comorbidities. Also, it may urge the food industries and services to adopt labelling GI values in nutrition facts labels of food products and to reformulate high GI foods to make them lower. However, this study protocol is subject to certain

limitations. First, the 2-week time frame of the intervention, especially in light of isoenergetic diet, may not be long enough to observe significant changes in liver fat and microbiome. Second, findings from small sample size may not be generalisable to all patients with NAFLD so will require replication in larger long-term RCT.

**Author affiliations**
[1]Nottingham Digestive Diseases Centre, School of Medicine, University of Nottingham, Nottingham, UK
[2]National Institute of Health Research (NIHR) Nottingham Biomedical Research Centre, Nottingham University Hospitals NHS Trust and the University of Nottingham, Nottingham, UK
[3]Al-Sabah Hospital, Ministry of Health, Civil Service Commission, Kuwait City, Kuwait
[4]School of Life Sciences, University of Nottingham, Nottingham, UK
[5]School of Medicine, University of Nottingham, Nottingham, UK
[6]Sir Peter Mansfield Imaging Centre, School of Physics and Astronomy, University of Nottingham, Nottingham, UK

**Acknowledgements** We thank Lu Ban for her assistance in the calculation of study sample size and acquisition of data. We also thank Andy Wragg and the NIHR Nottingham Biomedical Research Centre (BRC) Digestive Diseases Patient Advisory Group for their support in developing and design of the study. We are also grateful for support from liver and gastrointestinal disorder research team and for advice from Liz Simpson, Prarthana Thiagarajan, and Sally Cordon. The views expressed are those of the authors and not necessarily those of the NHS, the NIHR or the Department of Health.

**Contributors** AA-A, GA, JG, MT, AVa, AVi and SB conceived the study and designed the protocol. MT and AA-A contributed to diet planning. SB and PG involved in the discussion of the initial study design and subsequent outline and contributed to the planning of MRI/S data collection and analysis. AA-A drafted the manuscript and all authors revised the manuscript critically.

**Funding** This study is funded by University of Nottingham studentship to AA-A and National Institute for Health Research (NIHR) Nottingham Digestive Diseases Biomedical Research Unit and NIHR Nottingham Biomedical Research Centre (BRC-1215–20003) at the Nottingham University Hospitals NHS Trust and University of Nottingham. AA-A received academic scholarship from Ministry of Health, Civil Service Commission, State of Kuwait.

**Disclaimer** The study sponsor and funder will have no role in the design, collection, management, analysis, and interpretation of data; writing of the report; and the decision to submit the report for publication. The study sponsor and funder will have no ultimate authority over any of these activities.

**Competing interests** None declared.

**Patient and public involvement** Patients and/or the public were involved in the design, or conduct, or reporting, or dissemination plans of this research. Refer to the Methods section for further details.

**Patient consent for publication** Not applicable.

**Provenance and peer review** Not commissioned; externally peer reviewed.

**ORCID iD**
Amina Al-Awadi http://orcid.org/0000-0003-4333-7319

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
