## [Reviewer comments · BMJ Open]

ARTICLE DETAILS

TITLE (PROVISIONAL)	Effects of an iso-energetic low glycaemic index (GI) diet on liver fat accumulation and gut microbiota composition in patients with non- alcoholic fatty liver disease (NAFLD): A study protocol of an efficacy mechanism evaluation
AUTHORS	Al-Awadi, Amina; Grove, Jane; Taylor, Moira; Valdes, A; Vijay, Amrita; Bawden, Stephen; Gowland, Penny; Aithal, Guruprasad

VERSION 1 – REVIEW

REVIEWER	Utzschneider, Kristina University of Washington School of Medicine, Medicine
REVIEW RETURNED	05-Nov-2020

GENERAL COMMENTS	Overall, the proposal study protocol has many strengths and will rigorously assess in a small (n=16) set of adults the impact of dietary GI in the absence of weight change on liver fat. The microbiome assessment will add potential mechanistic insight, although the sample size is small. Strengths of the study protocol include: MRS is the most rigorous method to measure liver fat. The use of a cross-over design is a strength. The 4-week washout period should be adequate. The goal to prevent weight loss or gain will isolate out the effects of the GI alone, but will likely diminish the changes of seeing differences in liver fat with the interventions and will not be generalizable to the population. Comparison of CAP with MRS for response to intervention is a strength, as if highly correlated use of Fibroscan and CAP would be much more feasible for a larger intervention study. Use of food records and FFQs to assess usual diet habits is important and included. Weaknesses: 16S compositional analysis has the potential to miss functional microbiome changes that may be more meaningful. Production of SCFAs by specific “beneficial” bacteria in the gut is typically seen as beneficial within the gut itself. The authors should provide background justification for measuring SCFAs in the circulating plasma. Expand on the rationale for measuring GIP and GLP-1 in the fasting state. These increase in response to a meal and thus utility here is unclear. How are the stool samples collected – Are they being put into a container with RNA later? Or another microbiome kit? The public and patient involvement section can be shortened. The background is quite lengthy and could be shortened. Clean up the few grammatical errors in the manuscript.
--

REVIEWER	Jeznach-Steinhagen, Anna Medical University of Warsaw
REVIEW RETURNED	15-Nov-2020

GENERAL COMMENTS	If the results were presented in the table they would be clearer.
---

REVIEWER	Reddavid, Rosa IRCCS, Ambulatory of Clinical Nutrition
REVIEW RETURNED	07-Jan-2021

GENERAL COMMENTS	According to existing knowledge about microbiota colonization mechanisms, participants should avoid the assumption of probiotics at least for two weeks prior to the intervention period. In order to evaluate the effects of a Low Glycemic Index Diet, this one should be compared to a standard balanced diet for the general population, not to a High Glycemic Index Diet. Furthermore, it is known that a HGI diet is not a healthy choice, so this intervention can be considered not definitely ethic, even more in subjects with the eligibility criteria of the study (BMI >25, abdominal obesity and so on). Intervention time of two weeks seems to be really short to produce any kind of modification as in the gut microbiota, as in the liver fat content. The washout period of four weeks is appropriate. Finally, a Low Glycemic Index Diet is constituted by whole grains, pulses, fresh and not processed food (sourdough bread, legumes, eggs, fish and poultry, ecc), so the menu described is not the best choice to represent this dietetic pattern.
--

VERSION 1 – AUTHOR RESPONSE

Reviewer # 1

Strengths of the study protocol include:

MRS is the most rigorous method to measure liver fat. The use of a cross-over design is a strength. The 4-week washout period should be adequate. The goal to prevent weight loss or gain will isolate out the effects of the GI alone but will likely diminish the changes of seeing differences in liver fat with the interventions and will not be generalizable to the population. Comparison of CAP with MRS for response to intervention is a strength, as if highly correlated use of Fibroscan and CAP would be much more feasible for a larger intervention study. Use of food records and FFQs to assess usual diet habits is important and included.

Response 2: We agree with this comment. We are seeking funding for metagenomics analyses and will store extracted DNA for further analyses of microbiome functionality where consent permits. We have now added a sentence and reference in the introduction highlighting the potential benefits of SCFA generation [highlighted on page. 7, line 150-151, ref. 39]. In the methods section we have also now added background justification and references for measuring plasma SCFAs levels [highlighted on page. 18, lines 379-382, ref. 61 & 62].

Response 3: We agree this is an important point requiring clarification. In our study we will only measure GLP-1 in the fasting state since we will only be drawing blood once during the visit (when fasted) for participant comfort and practical/resource convenience reasons. We do not intend to quantify GIP in this study. We have, therefore, now inserted further explanation in the method section of revised manuscript justifying measurement of GLP-1 in the fasting state [highlighted on page 18, lines 374-379, ref. 60].

Response 4: Stool samples will be collected freshly on the study visit day by the participant and brought into the research centre along with icepack and immediately stored at 80°C. Stool samples will be processed using a standard commercial (Qiagen) kit for DNA extraction as described previously or equivalent. We will not use RNeasy. We have clarified this in the revision [highlighted on page.19, line 398-399, ref. 63].

Response 5: We have now shortened the PPI section [on page. 9].

Response 6: We have now shortened the background section from 1361 to 760 words.

Response 7: We apologize for not identifying and removing these errors before submission and appreciate that these have been highlighted. We have now corrected them.

Reviewer # 2:

Response 1: As the paper only describes the study design and methods of the study protocol there are no results presented. It is unclear how to address this comment – we could consider presenting Data Collection and Analyses of outcome measures in tabular format if this would be helpful.

Reviewer # 3:

Response 1: We appreciate that this is an important point to consider. Probiotics clearly influence gut microbes, evidence suggests that the gut microbiota changes quite rapidly in response to dietary changes. Previous studies with cross-over trial design have reported that a one-week washout period is acceptable to standardize gut microbiota and avoid any carry-over effects [Ref. 48 & 49]. Furthermore, research has shown that probiotic supplementation does not affect the composition of the gut microbiome [Kristensen NB, et al, 2016]. These short-term dietary changes are likely to have little impact on colonized populations but would be expected to influence the organisms which have not colonized. We therefore prefer to follow our HRA-approved protocol design as highlighted on page 15. We have now added the extra references in the methods section as justification [highlighted on page 15, lines 315-319].

Response 2: We agree with these comments and have fully considered it early in the planning and addressed these important points in the study design. Firstly, our study is an adaptation of a previous pilot study completed by members of our research group (Bawden, et al, 2017) that compared a 'standard' balanced LGI diet with a matched 'standard' balanced HGI diet which was originally conceived by Morgan, et al, (2012). Our study statistical design is based on the effect size reported in this earlier study so any changes to the intervention would invalidate the statistical planning. We also agree that HGI diet is not a healthy choice and would not be advisable for the study cohort, however, for that reason we are specifically selecting people who are already following a habitual HGI diet. In this way our study cohort is not representative of the 'general population' but rather of NAFLD patients who are known to consume a HGI diet [Ref 4]. We mention this in the introduction [highlighted on page. 6, lines 127-128]. To specifically avoid the additional excessive health risks that consuming HGI foods would place on any individual with NAFLD who follows a normal non HGI diet, we have stated that such people will be ineligible for the study [highlighted on page. 11, lines 238-240]. Participant's recruitment will be based on characterizing their habitual GI intake from 7-day food record

assessment. Those with a habitual lower GI diet intake (GI < 60) will be excluded from the study. Only participants who have a moderate to high GI diet intake (GI ≥ 60) will be recruited. By participating in this research project these individuals will be introduced to a LGI menu which may influence that eating behaviors and lead to future diet improvement. This addresses the ethical concern you have raised and allowed us to obtain ethical approval from national ethics committee.

Response 3: Our intervention is chosen as 2 weeks since the primary liver fat outcome proposed necessitates us to base the study on existing, published data [Ref. 21]. Although 2 weeks is a relatively short-time period we think it will be valuable to assess microbiome changes in this period (Deehan, et al, 2020) [Ref. 25]. We also believe that 2 week intervention period is acceptable to study the effect of LGI/HGI diet consumption on gut microbiota composition given the recent report indicating the changes in gut microbiota induced by diet within short time periods (David, et al, 2014; Wu, et al, 2016) [Ref. 29 & 27]. In addition, several studies have shown that dietary alterations can rapidly change the human gut microbiota composition in as little as 24 hours [Ref. 25-30]. We have highlighted this in the introduction [highlighted on page 6 & 7, lines 138-140].

Response 4: We accept this is a valid consideration, as in general processed foods are indeed higher in GI than fresh/wholegrains. However, we are following the definitions described on page 6 [lines 123-126] which were applied in the design of realistic UK diet menus described previously (Morgan et al., 2012) [ref. 41]. These menus were demonstrated to be HGI and LGI. We have taken these principles further in our previous pilot study and showed the diet menus achieve the study endpoints of reduced liver fat [ref. 21]. Based on this earlier publication, the statistical plan of our current protocol is calculated from the effect size reported [ref. 21], so we are aiming to stringently follow the diets described. The menus are very specifically designed to test if the GI and GL of diets have the same mechanistic effects on liver fat content in participants with NAFLD. Furthermore, these menus also consider the food safety issues related to providing food for participants that is going to be a low microbiological risk.

VERSION 2 – REVIEW

REVIEWER	Utzschneider, Kristina University of Washington School of Medicine, Medicine
REVIEW RETURNED	19-Mar-2021

GENERAL COMMENTS	Overall, the authors have responded to the reviewer comments. Summary: Strengths: change bullet 3 to read “dietary glycemic index” instead of “diet”. Others have examined the impact of other dietary interventions on liver fat using MRS. Limitations: This should be added: The 2 week time frame of the intervention, especially in light of isoenergetic diet, may not be long enough to observe significant changes in liver fat. Discussion: add limitations of the study
---

REVIEWER	Reddavid, Rosa IRCCS, Ambulatory of Clinical Nutrition
REVIEW RETURNED	13-Apr-2021

GENERAL COMMENTS	The study is innovative and of particular interest. But all the previous comments were not taken into account by the authors.
---

	As previously observed, two weeks of intervention are not enough time to change the microbiota composition. Three or four weeks would be a more adequate period of intervention. The low Glycemic index diet should be free from ready meals, and should be compared to a normocaloric balanced diet.
--	--

VERSION 2 – AUTHOR RESPONSE

Reviewer # 1

Comment 1: Summary: Strengths: change bullet 3 to read “dietary glycemic index” instead of “diet”. Others have examined the impact of other dietary interventions on liver fat using MRS. We have now corrected the strengths bullets in the summary section on page 4.

3

Comment 2: Limitations: This should be added: The 2 week time frame of the intervention, especially in light of isoenergetic diet, may not be long enough to observe significant changes in liver fat. we have added this point in the summary section on page 4.

Comment 3: Discussion: add limitations of the study We have added the study limitations at the end of discussion part [highlighted in page 27].

Reviewer # 3:

Comment 1:

The study is innovative and of particular interest. But all the previous comments were not taken into account by the authors. As previously observed, two weeks of intervention are not enough time to change the microbiota composition. Three or four weeks would be a more adequate period of intervention. We recognize the expertise of reviewer 3 in nutritional studies and did take this reviewer’s comments into consideration in the previous revision and responded by inserting the short-term duration as a limitation. However, we have previously explained that our intervention is specifically chosen as 2 weeks since the liver fat is the proposed primary outcome which necessitates us to base the study design directly on existing published data [Bawden et al, Ref. 21]. We have revised to emphasize this in the trial design section of methods and analysis on page 9. As reviewer 3 states, 2 weeks is a relatively short-time period, however we think it will be valuable to assess microbiome changes in this period as a secondary outcome of this study based on other studies (Deehan et al, 2020) [Ref. 25]. We also believe that 2-week intervention period is acceptable to study the effect of LGI/HGI diet consumption on gut microbiota composition given reports indicating changes in gut microbiota induced by diet within short time periods (David, et al, 2014; Wu, et al, 2016) [Ref. 29 & 27]. In addition, several studies have shown that dietary alterations can rapidly change the human gut microbiota composition in as little as 24 hours [Ref. 25-30]. We have highlighted this in the introduction [highlighted on page 6 & 7; lines 144-147].

Comment 2:

The low Glycemic index diet should be free from ready meals, and should be compared to a normocaloric balanced diet.

4

Again we appreciate this helpful expert insight but argue that this is not a relevant factor in our design of the current study as we intended to match closely the pre-designed, published dietary intervention. We have explained this comment raised by the reviewer previously that we are following the definitions described on page 6 [lines 130-133] which were applied in the design of realistic UK diet menus described previously (Morgan et al., 2012) [ref. 41]. Ready meals used in the diet menus were demonstrated to be HGI and LGI. We have taken these principles further in our previous pilot study and showed the diet menus achieve the study endpoints of reduced liver fat [ref. 21]. We have revised to

emphasize this in the trial design section of methods and analysis on page 9. Based on this previous publication which under-pins our protocol, the statistical plan of our current protocol is calculated from the effect sizes described [ref. 21], so we are aiming to stringently follow the diets described, indeed any changes to the diet would invalidate this calculation.

The menus are very specifically designed to test if the GI and GL of diets have the same mechanistic effects on liver fat content in participants with NAFLD. Furthermore, ready-to-eat meals have shown to be effective in lowering the glycemic response significantly in patients with diabetes [Manios, Y. et al, 2017; <https://pubmed.ncbi.nlm.nih.gov/26919992/>]. We respect the opinion of reviewer 3, however we decline to amend the previously-specified menu design. Our PubMed searches have been unable to identify any publications providing substantiation of the suggestion that ready-meal food is inappropriate for LGI diet. This food-type is widely used in dietary interventions with GI data available (Henry, CJ et al. Br J Nutrition 2005 94:922-30; doi: 10.1079/bjn20051594).